

# Circadian disruption and divergent microbiota acquisition under extended photoperiod regimens in chicken

Anne-Sophie Charlotte Hieke[1], Shawna Marie Hubert[1] and Giridhar Athrey[1,2]

[1] Poultry Science Department, Texas A&M University, College Station, TX, USA
[2] Faculty of Ecology and Evolutionary Biology, Texas A&M University, College Station, TX, USA

## ABSTRACT

The gut microbiota is crucial for metabolic homeostasis, immunity, growth and overall health, and it is recognized that early-life microbiota acquisition is a pivotal event for later-life health. Recent studies show that gut microbiota diversity and functional activity are synchronized with the host circadian rhythms in healthy individuals, and circadian disruption elicits dysbiosis in mammalian models. However, no studies have determined the associations between circadian disruption in early life, microbiota colonization, and the consequences for microbiota structure in birds. Chickens, as a major source of protein around the world, are one of the most important agricultural species, and their gut and metabolic health are significant concerns. The poultry industry routinely employs extended photoperiods (>18 h light) as a management tool, and their impacts on the chicken circadian, its role in gut microbiota acquisition in early life (first 3 weeks of life), and consequences for later life microbiota structure remain unknown. In this study, the objectives were to (a) characterize circadian activity under two different light regimes in layer chicken (12/12 h′ Light/Dark (LD) and 23/1 h LD), (b) characterize gut microbiota acquisition and composition in the first 4 weeks of life, (c) determine if gut microbiota oscillate in synchrony with the host circadian rhythm, and (d) to determine if fecal microbiota is representative of cecal microbiota in early life. Expression of clock genes (*clock*, *bmal1*, and *per2*) was assayed, and fecal and cecal microbiotas were characterized using 16S rRNA gene amplicon analyses from birds raised under two photoperiod treatments. Chickens raised under 12/12 LD photoperiods exhibited rhythmic clock gene activity, which was absent in birds raised under the extended (23/1 LD) photoperiod. There was differential microbiota acquisition under different photoperiod regimes in newly hatched chicks. Gut microbiota members showed a similar oscillating pattern as the host, but this association was not as strong as found in mammals. Finally, the fecal microbiota was found to be not representative of cecal microbiota membership and structure in young birds. This is one of the first studies to demonstrate the use of photoperiods to modulate microbiota acquisition in newly hatched chicks, and show their potential as a tool to promote the colonization of beneficial microorganisms.

Corresponding author
Giridhar Athrey,
giri.athrey@tamu.edu

Fecal microbiota, Cecal microbiota

## INTRODUCTION

Photoperiods and photo-intensity have played important roles in the success of domestic chickens as a globally important food source. Poultry products constitute a significant and growing proportion of global consumption (*Henchion et al., 2014*). Lighting has been one of the ubiquitous tools used to manage performance and welfare in broiler and layer production (*Ernst, Millam & Mather, 1987*; *Morris, 1967*). The use of photoperiods to stimulate egg-laying is one of the most important transformations in the commercial poultry industry, and in addition to modulating reproductive behavior (*Sharp, Quicke & Jansen, 1984*), lighting has been of interest in reducing cannibalism, optimizing feed intake and activity levels in modern poultry environments (*Ernst, Millam & Mather, 1987*; *Morris, 1967*). *Blokhuis (1983)* suggested that benefits of sleep in poultry are comparable to those in mammals, and several works have reported on the role of lighting for welfare (*Kristensen, 2008*; *Manser, 1996*; *Martrenchar, 1999*) and production (*Lewis & Morris, 1999*) in poultry. Whether photoperiods play the same role in modulating poultry health and homeostasis, as they do in mammals, remains unclear.

One of the key biological systems directly influenced by photoperiods is the circadian system, which has a well-documented role in influencing health. The circadian clock system is the central regulatory system that controls almost all aspects of an organism's behavior, physiology, and molecular function (*Cassone, 2015*; *Dawson et al., 2001*). The circadian is an evolutionarily conserved, hierarchically organized system with a master clock and peripheral clocks (*Bell-Pedersen et al., 2005*). For instance, circadian disruption is associated with a variety of metabolic, and immune disorders in mammals (*Archer et al., 2014*; *Buxton et al., 2012*; *Fonken et al., 2010*). In modern poultry rearing environments, extended photoperiods (EPs)—ranging from 14 to 23 h of light—are routinely used as a management practice (*Olanrewaju et al., 2006*). The impact of EPs has been addressed in poultry previously, but the existing literature has focused on balancing welfare and performance (*Deep et al., 2012*; *Schwean-Lardner, Fancher & Classen, 2012*). Recent studies of circadian disruption in humans have revealed multiple homeostatic processes regulated by the circadian system. These studies point to the critical role that circadian function plays in metabolic, immune, and musculoskeletal health, with a high relevance for livestock species (*Aoyama & Shibata, 2017*; *Di Cara & King-Jones, 2016*; *Ohta, Mitchell & McMahon, 2006*; *Shimizu, Yoshida & Minamino, 2016*; *Stothard et al., 2017*). However, we do not know how EPs influence the circadian system and clock-controlled processes, such as gut microbiota acquisition and gut health in poultry production, where lighting is a crucial management tool. A better characterization of these interactions is necessary to progress toward safe, secure and sustainable food for the future.

In birds, the master circadian clock is a tripartite system of pacemakers, including the pineal gland, the retinae, and the suprachiasmatic nucleus (SCN), which responds to

environmental cycles and photoperiods (*Cassone, 2014*; *Cassone & Westneat, 2012*). Peripheral clocks are found in almost all cells in the body and are synchronized with the master clock, ensuring specific day-night molecular processes that anticipate environmental and behavioral changes (*Albrecht, 2012*). At the molecular level, rhythmic expression of genes is controlled by a feedback loop that includes the positive elements (*clock* and *bmal1*), and the negative elements (*Period 2, Period 3, Cryptochrome 1, and Cryptochrome 2*) (*Cassone, 2014*). It has been shown in songbirds and galliformes (including chicken) that the rhythmic production of the pineal hormone melatonin entrains circadian rhythms. In mammals, the diurnal oscillations of circadian clock genes (*bmal1*, *clock*, *per2* etc.) and of clock-controlled genes (CCG) are an important indicator of health and homeostasis (*Mukherji et al., 2013*; *Thaiss et al., 2014*), whereas a disruption of normal circadian rhythms is associated with metabolic, and gut microbiota dysfunction (*Miyazaki et al., 2011*; *Shimizu, Yoshida & Minamino, 2016*). In birds, photoperiods directly or indirectly entrain circadian rhythms, with each of the three components (SCN, retinae, pineal) interacting to maintain master and peripheral clock rhythms (*Cassone, 2014*). As light can be perceived by both the pineal and retinal components of the avian clock, changes in light duration can render the avian circadian arrhythmic (*Cassone et al., 2008*). Evidence from avian studies on photoperiods and lighting intensity has demonstrated negative consequences for welfare traits (*Barbur et al., 2002*; *Prescott, Wathes & Jarvis, 2003*), as well as for eye development and function (*Barbur et al., 2002*; *Kristensen, 2008*; *Lauber, Shutze & Mcginnis, 1961*; *Nickla & Totonelly, 2016*). The expression of clock genes (*clock*, *bmal1*, and *bmal2*) in the pineal gland of the chicken has been demonstrated previously (*Kommedal, Csernus & Nagy, 2013*; *Nickla & Totonelly, 2016*; *Okano et al., 2001*), and while clock gene expression has been shown in peripheral tissues (*Chong et al., 2003*), the synchrony of peripheral rhythms with the master clock has not been demonstrated. In poultry species, clock gene expression (*bmal1*, *per3*) in the pineal gland (*Turkowska et al., 2014*), and melatonin production (*Kommedal, Csernus & Nagy, 2013*) do not display rhythmicity under constant dark or light conditions.

Recent work has revealed that gut microbiota show rhythmic oscillations in synchrony with the host circadian clock (*Thaiss et al., 2014*). In most vertebrates, including chicken, commensal microorganisms colonize the gastrointestinal tract (*Pritchard, 1972*; *Salanitro, Fairchilds & Zgornicki, 1974*; *Waite & Taylor, 2014*), forming the gut microbiota community. Early studies such as *Apajalahti, Kettunen & Graham (2004)* showed that the chicken gastrointestinal tract is colonized rapidly in the first days of life, but the study did not illuminate the membership of this early community. In terms of diversity and complexity, and the immune maturation it elicits, it has been shown that acquisition of new taxa continued up to and beyond day 19 (*Crhanova et al., 2011*). This data supports the view that the early life microbiota acquisition is crucial for the establishment of a stable microbiota in later life (*Stanley et al., 2013*). The diversity of microbiota, acquired early in life, can be critical for the regulation of immune and metabolic health in vertebrates (*Cox et al., 2014*; *Lee et al., 2013*; *Moloney et al., 2014*; *Subramanian et al., 2015*; *Thaiss et al., 2014*) and also in chicken (*Crhanova et al., 2011*; *Kogut, 2013*;

*Stanley, Hughes & Moore, 2014*). However, no studies to date have characterized this relationship between circadian and the microbiota in birds. In domestic chicken, these associations take on special significance; the EPs used in poultry production systems likely disrupt normal circadian rhythms, and influence the normal acquisition of microbiota, and establishment of stable communities. Additionally, as the poultry industry transitions to antibiotic-free production, there is an urgent need to identify economical solutions for promoting gut health. If gut microbiota structure and membership can be influenced by photoperiods in early life, this approach can become a potentially valuable, and economical approach to manage gut and metabolic health in poultry.

One common feature of most commercial production systems is the lighting regimens that newly hatched chicks are reared under. Both broiler and layer chicks are started at 20–23 h of continuous light during the first few weeks of their life. While broilers are maintained at EPs for the entirety of their life (6–7 weeks), layer chicks follow a varying photoperiod regimen until sexual maturity. In both cases, chicks experience 20+ h of continuous lighting for the first few weeks of life. This early-life period also overlaps with a crucial window for the acquisition of the gut microbiota, which in turn is linked with later life metabolic and immune homeostasis. It is being increasingly recognized that early life microbiota acquisition determines the later life microbiota structure and diversity.

In this study, we investigated the relationship between EPs, host circadian oscillations and the gut microbiota acquisition under two photoperiod regimens (12/12 Light/Dark (LD) and 23/1 LD). Additionally, this study also tracked the early life cecal microbiota in the first 3 weeks of life to determine whether and when cecal microbiota communities diverge under different photoperiods. *Wang et al. (2018)* assessed this phenomenon in 20-week-old broilers, but the present study focused specifically on microbiota acquisition in newly hatched chicks. Finally, we compared fecal and cecal microbiotas in the first 3 weeks (period of circadian entrainment, and microbiota establishment) to answer whether the fecal microbiota are representative of early life cecal microbiota. Previous studies have showed that fecal microbiota in chicken is not representative of cecal microbiota in older birds, but in this study we focused on early life (*Stanley et al., 2015*). We speculated that if early life fecal microbiota is informative about early cecal microbiota, it would enable longitudinal studies where birds can be sampled repeatedly.

## MATERIALS AND METHODS

### Animal ethics statement

All animal work was conducted in accordance with national and international guidelines for animal welfare. The animal trials were approved and monitored by the Institutional Animal Care and Use Committee of Texas A&M University (Assurance Number 2016-0064).

### Animals and experimental design

All birds used in the study were female Hy-Line Brown Layers (*Gallus gallus domesticus*). A total of 80 hatch-day chicks were obtained from a local hatchery and transported to the Texas A&M Poultry Research and Education Center in College Station, Texas.

A total of 40 chicks were randomly assigned to one of two treatments, and then moved into one of two identical environmental chambers with independent lighting controls. Within each chamber, 20 chicks were placed into one of two brooder cages. Each environmental chamber was set to one of the photoperiod treatments—normal photoperiod (NP) of 12 h of light and 12 h of darkness (12/12 LD), with lights-on at 06:00 h, and EP treatment of 23 h L and 1 h D (23/1 LD), with lights-off from 05:00 to 06:00 h. Following the convention from circadian studies, Zeitgeber Time 0 (ZT0) was defined as the time of lights-on (0600 h). A total of 40 birds were raised under each photoperiod. Except for the photoperiod exposure, the experimental birds experienced identical conditions, and had ad libitum access to feed and water. Chicks were reared on a pullet diet comprising 17% crude protein, with an energy concentration of 2,800 kcal metabolizable energy per kg. Temperature-controlled experimental rooms were maintained at $32 \pm 2\ °C$ for the first week and then decreased by ca. 2–3 °C per week down to 23 °C, following the Hy-Line management guide.

## Sample collection

For birds raised under each photoperiod, we monitored early-life cecal and fecal microbiotas for the first 19 days of life (entrainment period), followed by 2 days of circadian sampling (19–21 days old). To monitor the cecal microbiota during the entrainment period (Day 1–18), chicks were euthanized every other day at ZT1 (12:00 h) starting on Day 4 ($n = 1$ individual/treatment/day) and the cecal content was collected and stored as described below. In addition, two fecal samples were collected every day (Day 1–20) from both groups at ZT1. These fecal samples were depositions of individual birds. To ensure collection of fecal samples deposited close to ZT1, fecal trays were lined with clean lab bench paper, which was replaced after every sampling event, and only fresh fecal samples were collected. Fecal samples were transported to the laboratory on ice and stored at −80 °C until further processing.

At the end of the entrainment period (19 days), two birds were randomly selected and euthanized at 6-h intervals to characterize circadian oscillations. Individual birds were euthanized by exposure to 5 min of $CO_2$ followed by cervical dislocation. Two birds from each photoperiod treatment were sampled this way every 6-h (two individuals/treatment/time point) over a 48-h period, starting at ZT0 (nine time points, two birds each at each time point, total 18 per treatment). For collections in the dark period (NP), birds were taken in the dark using only an infrared lamp to avoid light exposure, and placed in a dark container which was used as the euthanasia chamber. Tissue samples (brain, ceca, cecal content) were collected within 30 min of euthanasia and immediately placed into RNALater (Qiagen, Hilden, Germany) in 1:5 ratio. Both ceca were removed and the bottom tips were separated. Cecal content from each cecum was then gently squeezed into a sterile collection tube to obtain enough cecal content for downstream analyses. As birds from both treatments had to be sampled at exactly the same times, four personnel simultaneously performed identical steps from euthanasia to tissue collection, within 30 min post-mortem. Following the dissections, each tissue sample was stored in separate tubes at 4 °C for at least 24-h to ensure complete penetration of

RNALater. Following the removal of RNALater, the samples were stored at −80 °C. A total of 18 individual samples were collected (nine time points × two birds per time point) for each photoperiod treatment. These 18 samples per treatment were used for microbiota community comparisons between the normal and EPs.

## DNA/RNA isolation and gene expression analyses

Brain and ceca tissue samples were homogenized in Trizol reagent (Invitrogen, Carlsbad, CA, USA) using a hand-held Tissuemiser (Fisher Scientific, Hampton, NH, USA) and total RNA was extracted according to the manufacturer's instructions. Tissue samples were collected for expression analysis from two individuals at each of nine time points over a 48-h period (6-h intervals), for each photoperiod treatment. 100 ng of total RNA were used to generate cDNA using the SuperScript VILO MasterMix RT-PCR kit (Invitrogen, Carlsbad, CA, USA). RealTime PCR was performed using gene-specific primers (Integrated DNA Technologies, Coralville, IA, USA) and PowerUp SYBR Green Master Mix (Applied Biosystems, Foster City, CA, USA) on a 7900HT Fast Real-Time PCR System (Applied Biosystems, Foster City, CA, USA), and using Actin as the housekeeping gene. PCR conditions were 50 °C for 2 min, 95 °C for 2 min, followed by 40 cycles of 95 °C for 15 s and 57 °C for 1 min. The primers used for qPCR of clock genes were the same as reported in *Okano et al. (2001)*. Primer sequences are shown in Table S1.

## Microbiota analysis

DNA from cecal content and fecal samples was extracted using the MoBio PowerFecal (Qiagen, Hilden, Germany) kit according to the manufacturer's instructions. Each sample was initially homogenized using a BioSpec Mini-Beadbeater (BioSpec Products, Bartlesville, OK, USA). A total of 20 ng of purified DNA were used for PCR amplification of bacterial 16S rRNA gene sequences, using Q5® High-Fidelity DNA polymerase (NEBNext® High-Fidelity 2X PCR Master Mix, New England BioLabs, Ipswich, MA, USA). We used a 15-cycle PCR to first amplify the 16S rRNA gene sequences (in triplicate) followed by seven-cycle PCR to add the Illumina barcodes. The V4 primer pair was specifically chosen to avoid amplification of eukaryotic 18S rRNA gene sequences (Hyb515F_rRNA: 5′-TCGTCGGCAGCGTCAGATGTGTATAAGAGACAGGTGYCAG CMGCCGCGGTA -3′, Hyb806R_rRNA: 3′-TAATCTWTGGGVHCATCAGGG ACAGAGAATATGTGTAGAGGCTCGGGTGCTCTG-5′) (*Wang & Qian, 2009*). A non-template negative control (blank) was included in the amplification step, and visualized on an agarose gel. As no bands were observed for the negative controls, the blanks were not included in the sequencing library. Barcoded amplicons were cleaned up using Ampure beads (Beckman Coulter, Brea, CA, USA). Library preparation and sequencing was performed at Genome Sequencing and Analysis Facility (GSAF, University of Texas, Austin, TX, USA). Amplicons were sequenced in 2 × 250 bp paired-end mode on an Illumina MiSeq platform (Illumina, San Diego, CA, USA). Reads were processed using the Mothur software, version 1.38 (*Schloss, 2009*). Briefly, paired-end reads were joined using the make.contigs command. Sequences of incorrect length and with ambiguous base calls were removed using the screen.seqs command. The remaining

sequences were aligned against the SILVA database (release 123) (*Quast et al., 2013*) using the NAST algorithm (*DeSantis et al., 2006*) and screened for homopolymers greater than eight bases. Chimeras were removed with UCHIME (*Edgar et al., 2011*) and sequences were classified against the SILVA taxonomy (*Yilmaz et al., 2014*) using the Bayesian classifier (*Wang et al., 2007*). Sequences that classified to Eukaryota, Archaea, chloroplast, mitochondria, or unknown were removed from the data set. Sequences were clustered into operational taxonomic units (OTUs) of 97% sequence similarity using the average neighbor algorithm (default). Rarefaction curves for the observed number of OTUs were generated in Mothur using 1,000 randomizations. Weighted and Unweighted Unifrac analyses were also performed using the Mothur software. α diversity and the impact of other variables (photoperiod, sample type, and age) on community differences was analyzed and compared using the Phyloseq (version 1.14.0) (*McMurdie & Holmes, 2013*) and vegan (version 2.4-2) (*Oksanen et al., 2017*) packages in the R software environment (*R Development Core Team, 2012*). The raw sequence data reported in this study are available for download from FigShare (https://doi.org/10.6084/m9.figshare.6938249).

As singletons and low abundance OTUs can inflate measures of diversity, and bias community analysis (*Kunin et al., 2010*; *Schloss, Gevers & Westcott, 2011*; *Zhan et al., 2014*) singletons and low abundance OTUs were filtered out. The total dataset was filtered at two thresholds recommended in the Phyloseq manual—namely $10^{-5}$ (0.01%) and a more stringent $10^{-3}$ (1%) threshold, based on the mean abundance across samples. We considered these filtered data thresholds to be more biologically relevant, especially from the point of detecting taxa that oscillate rhythmically across time points. For taxa occurring at very low abundance, it may be difficult to distinguish presence-absence resulting from low biological occurrence vs. an oscillating pattern generated due to circadian rhythmicity in microbial abundance. Our inferences and discussion are based on the 0.01% threshold, but we report 1% threshold data for comparison.

Principal coordinates analysis (PCoA) and non-metric multidimensional scaling plots were created in R. Permutational multivariate analysis of variance (PERMANOVA) with linear model fitting (*Anderson, 2001*; *McArdle & Anderson, 2001*) using the "Adonis" function in the vegan package was performed to test how well the groupings, based on the metadata factors, accounted for the variation between the samples. Statistical tests of α and β diversity (PERMANOVA, metastats, LEfSe) between the two photoperiods were based on 18 replicates per treatment. All other statistical tests were performed in R. We investigated the directionality and extent of differences in microbiota between the two photoperiod treatments, using the program Metastats (within Mothur), and the non-parametric linear discriminant analysis (LDA) tool LEfSe. The latter approach is used to detect biomarkers that differ between two or more phenotypes in a metagenomic context. The non-parametric approaches are considered more robust to violations of normality that is typical of smaller datasets such as the current study.

## Analysis of circadian oscillations

Gene expression values and microbial abundance data were both analyzed for rhythmic oscillations using the JTK_cycle test (*Hughes, Hogenesch & Kornacker, 2010*). JTK_Cycle is

a program that performs the Jonckheere-Terpstra-Kendall nonparametric test for detecting patterns and ordering across independent groups. In this context, the program tests for rhythmic changes in the length of circadian period (the amount of time between a recurring event), and the phase (the time of peak activity). The implementation of Kendalls' Tau is known to reduce the impact of outliers, and hence provides a more robust detection of periods and phases. Furthermore, this program has been shown to be less prone to false positives compared to other commonly used tests for circadian rhythms (*Hughes, Hogenesch & Kornacker, 2010*). For the analysis of rhythmic oscillations and their amplitudes we used a window of 24–36-h for the detection of circadian periodicity and phase. Genes were considered to display rhythmicity at a significance threshold of BH.Q < 0.05 (Benjamini–Hochberg *Q*-value). The BH.Q value is a more stringent threshold for significance as it protects against false positives. The dataset for analyses of both gene expression and microbiota profiles comprised 18 samples for each photoperiod treatment (nine time points × two birds per time point). While two replicates per time point is low for circadian gene expression studies, in this case we deemed these numbers to be sufficient given that our circadian gene expression analysis was intended to confirm a well-documented phenomenon. The community analyses between photoperiod treatments were based on 18 individuals per treatment (36 total).

## RESULTS

### Absence of circadian rhythms under extended photoperiods

Circadian oscillations, and their corresponding period and phase, were analyzed using the gene expression data for three clock genes (*clock, bmal1, per2*) from the time-series experiment. JTK_Cycle analysis showed that all three assayed genes oscillated with significant 24-h rhythms in the brains of chicks entrained to the NP (12/12 h LD), whereas such rhythms were absent in the brains of the chicks entrained to the EP (23/1 h LD) (Fig. 1). In the brain, *Clock* and *bmal1* gene expression peaked toward the beginning of the dark phase (scotophase), and was at its lowest expression toward the start of the light phase (photophase). *Per2* mRNA levels peaked at the end of the scotophase, and were lowest toward the end of the photophase. These genes displayed a significant rhythmic oscillation based on the JTK-Cycle test, with BH.Q values < 0.0003 for all three genes. In contrast, gene expression levels in chick brains exposed to the EP did not show distinct oscillation patterns. *Clock* and *per2* mRNA levels did not oscillate at all (BH.Q > 0.05), whereas *bmal1* mRNA levels did show a weak oscillation pattern, reaching lowest expression during the 1-h scotophase. *Bmal1* was the only gene showing oscillation detectable by JTK_Cycle (BH.Q = 0.038) in the EP treatment. These results show that chicken raised under a NP treatment have a functioning circadian rhythm as displayed by three major clock genes, whereas chicken raised under EP treatment do not show comparable rhythms.

Clock gene (*clock, bmal1, per2*) expression levels in the ceca followed the same rhythmic pattern as the brain, but with a delayed phase, where the peak expression is shifted forward a few hours (Fig. 2). As in the brain tissue, these three genes showed significant oscillation based on JTK_Cycle (BH.Q < 0.05) in the NP treatment. However, in the

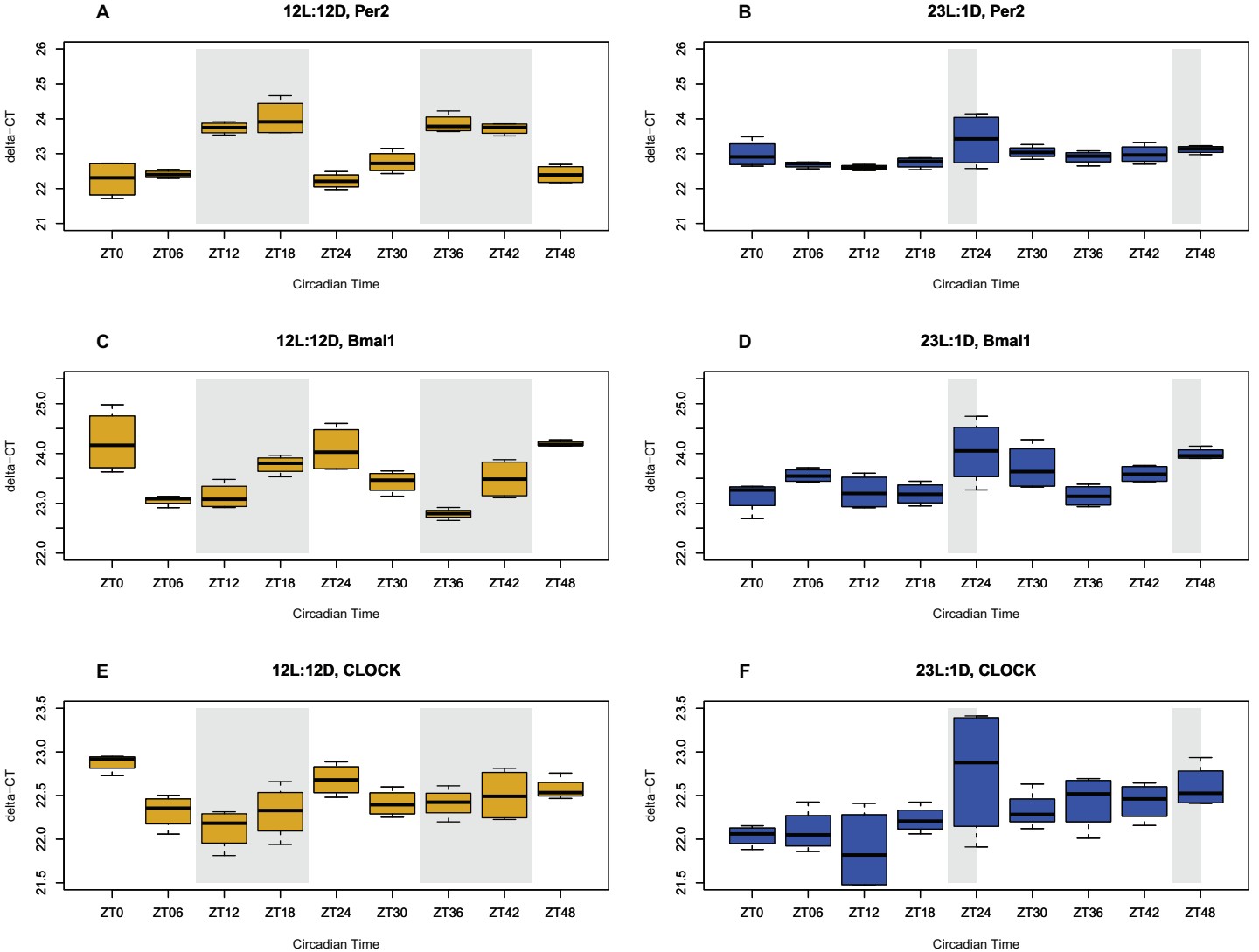

**Figure 1 Circadian gene expression profiles in the brain.** Expression of clock genes in the brain tissue of chicks entrained to either normal (12L:12D) (yellow) or extended photoperiods (23L:1D) (blue), measured with qPCR. The shaded areas represent the hours of darkness. *X*-axis give the time scale in zeitgeber (ZT) over the 48 h sampling period, and the *Y*-axis shows the level of gene expression in delta-CT. Expression patterns of the *Per2*, *Bmal1*, and *Clock* genes in NP (A, C, E respectively), and in EP (B, D, F respectively) are shown.

EP treatment, none of these genes showed a significant oscillation pattern (BH.Q > 0.05). These results show that the peripheral clock in the ceca is synchronized with the clock in the brain tissue and also oscillates in a 24-h rhythm under the NP but not in the EP treatment.

## Different photoperiods promote differential microbiota membership and structure

Amplicon sequencing resulted in 495,572 sequences, of which 442,177 sequences were retained after quality filtering (wrong length and ambiguous base calls). Sequence counts per sample averaged 13,614-paired reads. Following the analysis of microbiota using

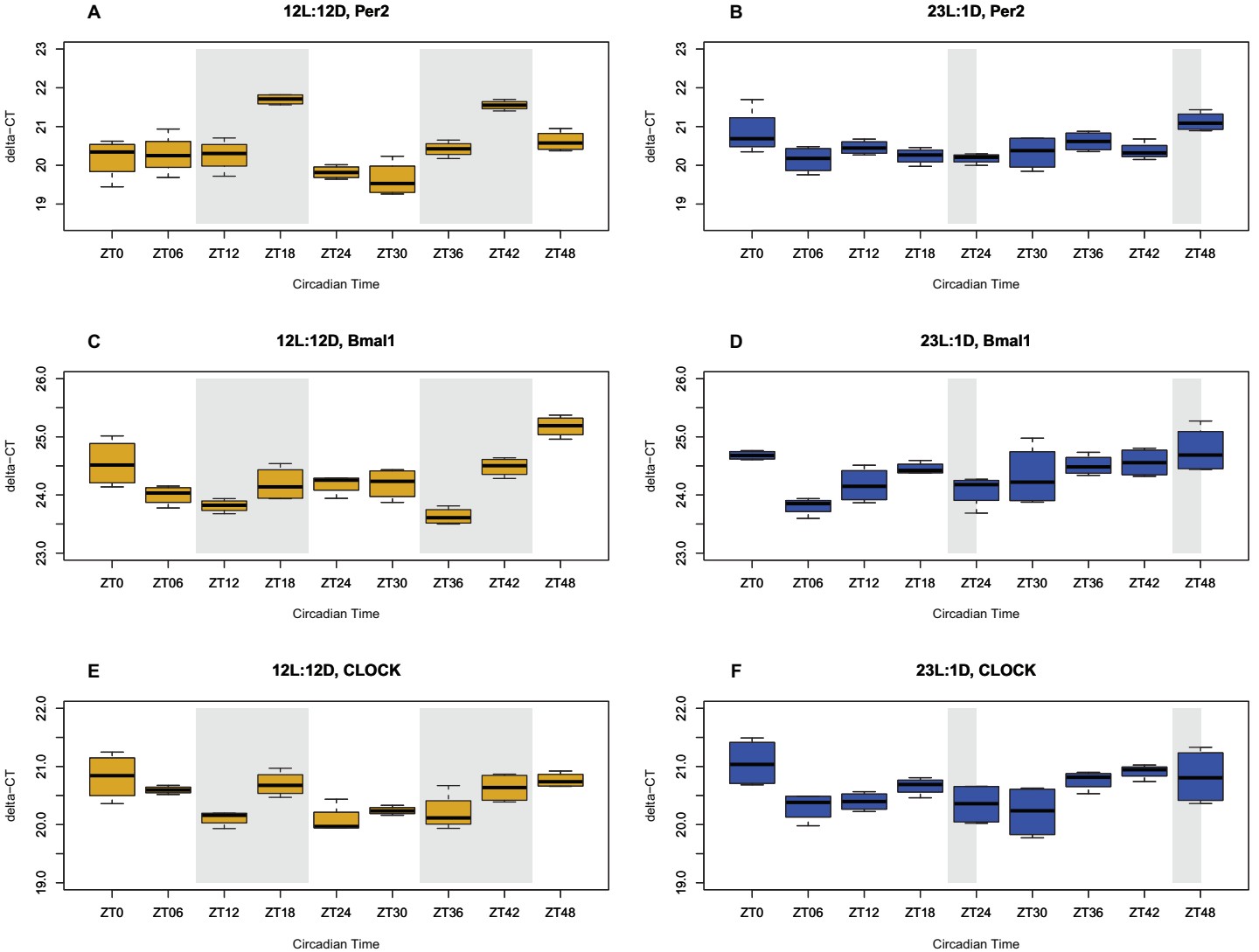

**Figure 2 Circadian gene expression profiles in the ceca.** Expression of clock genes in the ceca of chicks entrained to either normal (12L:12D) (yellow) or extended photoperiods (23L:1D) (blue), measured with qPCR. The shaded areas represent the hours of darkness. *X*-axis shows the time scale in zeitgeber (ZT) over the 48 h sampling period, and the *Y*-axis shows the level of gene expression in delta-CT. Expression patterns of the *Per2*, *Bmal1*, and *Clock* genes in NP (A, C, E respectively), and in EP (Figure B, D, F respectively) are shown. Error bars are standard errors.

the Mothur pipeline, a total of 843 OTUs were observed in the entire data set. The 843 OTUs were classified into 19 phyla, 89 families, and 118 genera. Among these, 595 OTUs were classified into 14 phyla, 58 families, and 94 genera in the NP treatment. In the EP treatment, we observed 646 OTUs that were classified into 18 phyla, 75 families, and 100 genera.

Above the 0.01% threshold, 382 OTUs (45% of the original 843 OTUs) were retained that were classified into 10 phyla, 36 families, and 69 genera. At this abundance threshold, 14 and 11 OTUs were found exclusively in the NP and EP treatments, respectively. A list of these OTUs can be found in the supplementary data (Table S1). At the 1% threshold, a total of 190 OTUs (23% of the original 843 OTUs) were retained

that were classified into seven phyla, 20 families, and 43 genera. For the NP treatment, the dominant phylum was *Firmicutes* (94.2%), followed by *Tenericutes* (1.3%), *Actinobacteria* (0.65%), and *Proteobacteria* (0.14%). For the EP, the dominant phylum was also *Firmicutes* (90.89%), followed by *Bacteroidetes* (2.92%), *Tenericutes* (1.19%), *Actinobacteria* (0.63%), and *Proteobacteria* (0.15%). At the genus level (>1%), the NP was dominated by *Faecalibacterium* (24.5%), followed by *Lachnoclostridium* (8.9%), *Ruminococcaceae*_UCG-014 (7.1%), *Anaerotruncus* (4.1%), and *Lactobacillus* (3.7%). The EP treatment was also dominated by *Faecalibacterium* (31.3%), followed by *Ruminococcaceae*_UCG-014 (8.1%), *Lachnoclostridium* (7.8%), *Anaerotruncus* (4.0%), and *Alistipes* (2.9%). Stacked bar plots depicting all the classified genera above 1% relative abundance for both the NP and EP treatments are shown in Fig. 3. Similar plots for family level classifications are given in Fig. S1. The two photoperiods shared 129 OTUs (80.1%) and 18 (11.2%) and 14 (8.7%) OTUs were unique to the normal and EPs, respectively. A list of unique OTUs for each photoperiod is presented in Table 1.

Next, the OTU tables were used to estimate α and β diversity. The PCoA plot showed that the two communities do not cluster completely independently of each other, and show some overlap (Fig. 4). Additional PCoA plot with the second and third components are shown in Fig. S2, and a network plot based on distances is shown in Fig. S3. However, α diversity estimates using Mann–Whitney U-tests were significantly higher (Z-Score = −1.91, *P* = 0.02) for the NP group across different estimators (Chao, Simpson, Inverse Simpson), showing that NP photoperiods supported a higher overall microbial diversity (Fig. 5). This pattern was consistent during the entrainment period (first 3 weeks) (Fig. 5A), and when looking only at the samples collected during the circadian sampling (Fig. 5B).

To compare the microbial community between treatments (β diversity), we used PERMANOVA, parsimony (clustering within tree), as well as Weighted and Unweighted Unifrac analyses. The PERMANOVA analysis on the Bray-Curtis distances revealed that the cecal gut microbiota communities were significantly different for the two photoperiods (*P* = 0.002). Similarly, β diversity between the NP and EP groups were found to be significantly different using the parsimony (*P* = 0.034), unweighted UniFrac (*P* < 0.001), as well as weighted UniFrac (*P* < 0.001) approaches. The weighted and unweighted UniFrac analyses both show that membership and structure of the microbiota communities were different between the photoperiod treatments.

Metastats analysis showed that 62 taxa (16% of total) occurred at significantly different abundance (*P* < 0.05) between the two light treatments. The LEfSe analysis showed that 33 total taxa were differentially enriched between the two treatments, of which 26 were enriched in NP and seven were enriched in EP treatments, respectively, but several of these taxa are not classified beyond the genus level. The top enriched taxa by effect size (LDA score) were *Rikenellaceae* (*Alistipes*), *Lachnospiraceae*, and Ruminococcaceae in EP. In the NP treatment, the top enriched taxa were *Lachnospiraceae*, *Ruminococcaceae*,

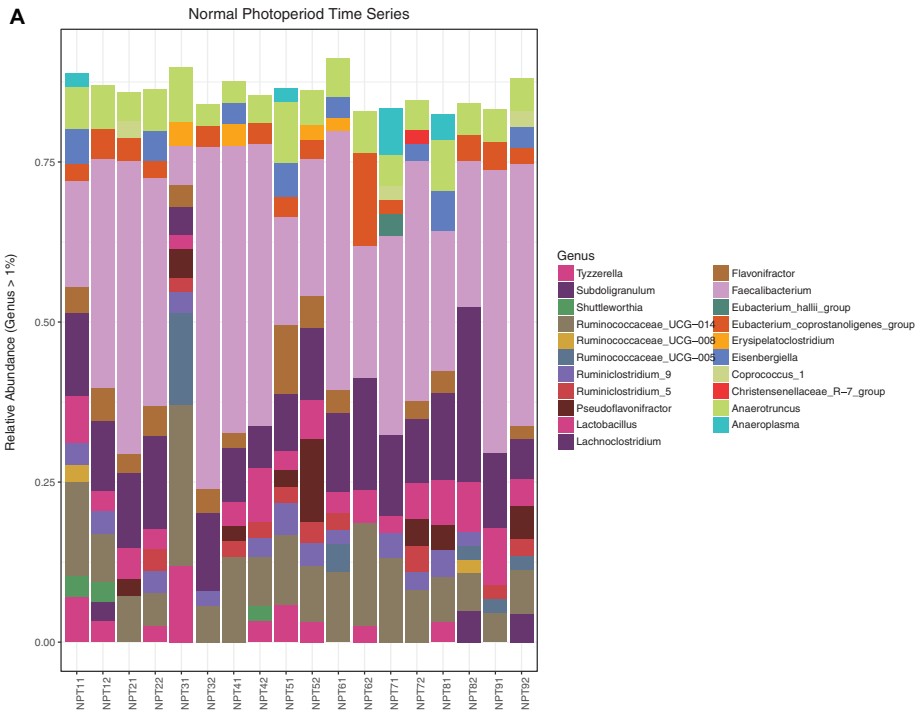

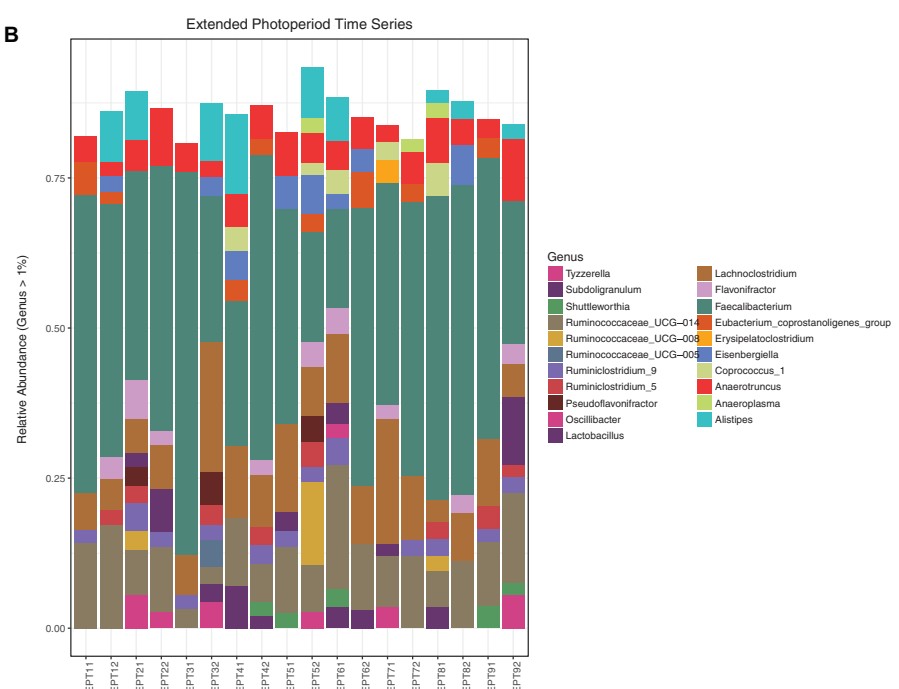

**Figure 3 Column plots of relative abundance at the genus level.** Relative abundance (>1%) at the taxonomic genus level depicting the diversity of cecal microbial communities in chicks entrained to the normal photoperiod (A) and the extended photoperiod (B). NP and EP labels correspond to birds sampled from either treatment, with two replicates per time point. Samples were taken at 6-h intervals over a 48-h period from Day 19–21.

**Table 1 Taxa that were found either in the normal or the extended photoperiod treatments only.**

| Phylum | Class | Order | Family | Genus |
|---|---|---|---|---|
| **Taxa that were found uniquely in the normal photoperiod treatment** | | | | |
| Firmicutes | Clostridia | Clostridiales | Ruminococcaceae | Faecalibacterium |
| Firmicutes | Clostridia | Clostridiales | Lachnospiraceae | Tyzzerella |
| Firmicutes | Clostridia | Clostridiales | Peptostreptococcaceae | NA |
| Firmicutes | Clostridia | Clostridiales | Lachnospiraceae | NA |
| Firmicutes | Clostridia | Clostridiales | Lachnospiraceae | Lachnoclostridium |
| Firmicutes | Clostridia | Clostridiales | Ruminococcaceae | Ruminococcaceae_UCG-014 |
| Firmicutes | Clostridia | Clostridiales | Ruminococcaceae | NA |
| Firmicutes | Clostridia | Clostridiales | Lachnospiraceae | NA |
| Firmicutes | Clostridia | Clostridiales | Lachnospiraceae | NA |
| Firmicutes | Clostridia | Clostridiales | Lachnospiraceae | Lachnospiraceae_NC2004_group |
| Firmicutes | Erysipelotrichia | Erysipelotrichales | Erysipelotrichaceae | NA |
| Firmicutes | Clostridia | Clostridiales | Lachnospiraceae | NA |
| Firmicutes | Clostridia | Clostridiales | Lachnospiraceae | NA |
| Firmicutes | Clostridia | Clostridiales | Ruminococcaceae | Ruminococcaceae_UCG-010 |
| Firmicutes | Clostridia | Clostridiales | Christensenellaceae | Christensenellaceae_R-7_group |
| Firmicutes | Clostridia | Clostridiales | Lachnospiraceae | NA |
| Firmicutes | Clostridia | Clostridiales | Lachnospiraceae | NA |
| Firmicutes | Clostridia | Clostridiales | Christensenellaceae | Christensenellaceae_R-7_group |
| **Taxa that were found uniquely in the extended photoperiod treatment** | | | | |
| Firmicutes | Clostridia | Clostridiales | Clostridiaceae_1 | Candidatus_Arthromitus |
| Bacteroidetes | Bacteroidia | Bacteroidales | Rikenellaceae | Alistipes |
| Bacteria_unclassified | Bacteria_unclassified | Bacteria_unclassified | Bacteria_unclassified | NA |
| Firmicutes | Clostridia | Clostridiales | Clostridiales_vadinBB60_group | NA |
| Firmicutes | Clostridia | Clostridiales | Clostridiales_unclassified | NA |
| Tenericutes | Mollicutes | Mollicutes_RF9 | Mollicutes_RF9_unclassified | NA |
| Firmicutes | Clostridia | Clostridiales | Ruminococcaceae | Ruminiclostridium_5 |
| Tenericutes | Mollicutes | Mollicutes_RF9 | Mollicutes_RF9_unclassified | NA |
| Tenericutes | Mollicutes | Mollicutes_RF9 | Mollicutes_RF9_unclassified | NA |
| Firmicutes | Clostridia | Clostridiales | Lachnospiraceae | NA |
| Bacteria_unclassified | Bacteria_unclassified | Bacteria_unclassified | Bacteria_unclassified | NA |
| Firmicutes | Clostridia | Clostridiales | Ruminococcaceae | Ruminiclostridium_9 |
| Firmicutes | Clostridia | Clostridiales | Lachnospiraceae | NA |
| Firmicutes | Clostridia | Clostridiales | Ruminococcaceae | Anaerotruncus |

**Note:**
List of unique taxa (>1% relative abundance) for the normal and extended photoperiods.

and *Lactobacillaceae (Lactobacillus spp.)* (Fig. 6). Of the top 10 most enriched taxa in the NP group, three were of the genus *Lactobacillus*.

## Rapid cecal microbiota divergence under different photoperiods

To understand how long after entrainment under different photoperiods the cecal microbiota communities diverge, median α diversity indices over the first 3 weeks were

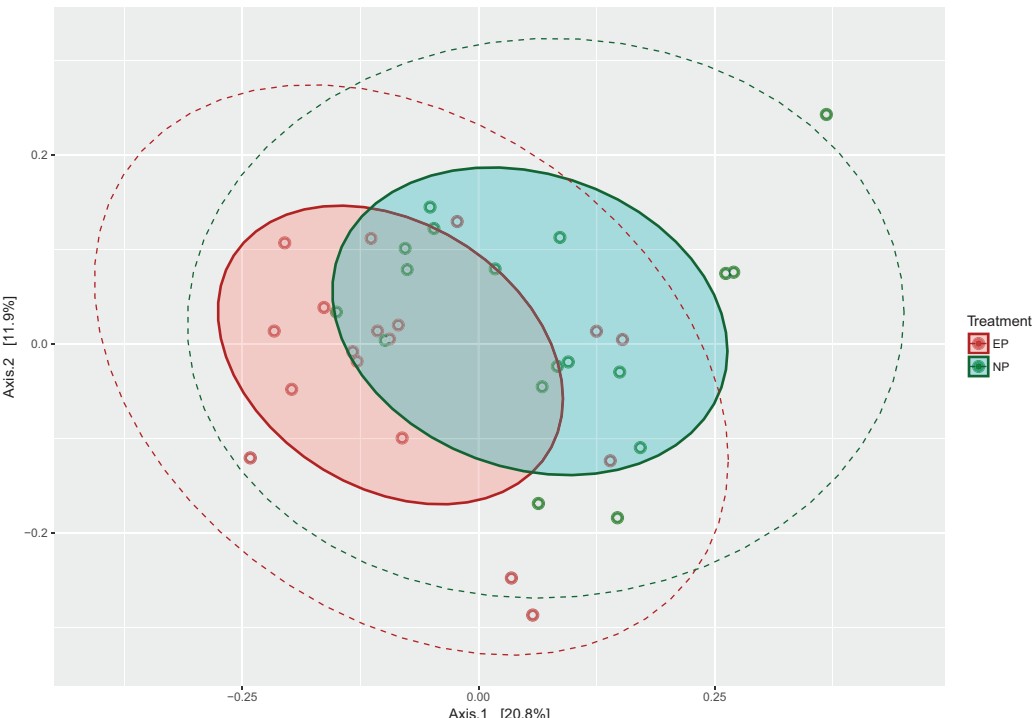

**Figure 4 PCoA plot of microbiota communities from the two photoperiods.** Principal Coordinate Analysis (PCoA) plot of cecal microbial communities entrained under normal photoperiods (NP) and extended photoperiods (EP). Solid shaded ellipses around colored points show the 90% Euclidean distance from the center, whereas dashed lines show the 95% normal distribution span.

compared (Fig. 5). Cecal microbiota during the entrainment period (first 20 days), grouped by weeks since hatch (weeks 1, 2, 3), showed that α diversity increased linearly in both treatments, but there was weak correlation between the two photoperiods ($R^2$ = 0.58, $P$ = 0.10). Overall, the EP group had lower median α diversity values compared to the NP treatment, but these differences were not statistically significant for the whole group. The non-parametric Mann–Whitney U test, showed that α diversity values were statistically different in the second week ($Z$-score = −2.28, $P$ = 0.013), and in the third week ($Z$-score = −1.69, $P$ = 0.045). Median α diversity for the first week compared using Chi-square goodness-of-fit test (due to low replication) was also significantly different ($\chi^2$ = 52.61, d$f$ = 1, $P$ < 0.001). Comparisons of β diversity using AMOVA and PERMANOVA were not meaningful, owing to the small sample sizes. However, Metastats analysis showed an increasing number of differentially abundant taxa with every passing week. There were five (1.3% of total), 18 (4.7% of total), and 23 (6% of total) taxa found at significantly different abundances in Week 1, Week 2, and Week 3, respectively, between the two photoperiod treatments. In summary, microbiota structure appears to differentiate starting within the first few days of life under different photoperiods, but greater replication is necessary to confirm this finding. Additionally, cage or room effects also need to be considered in future studies.

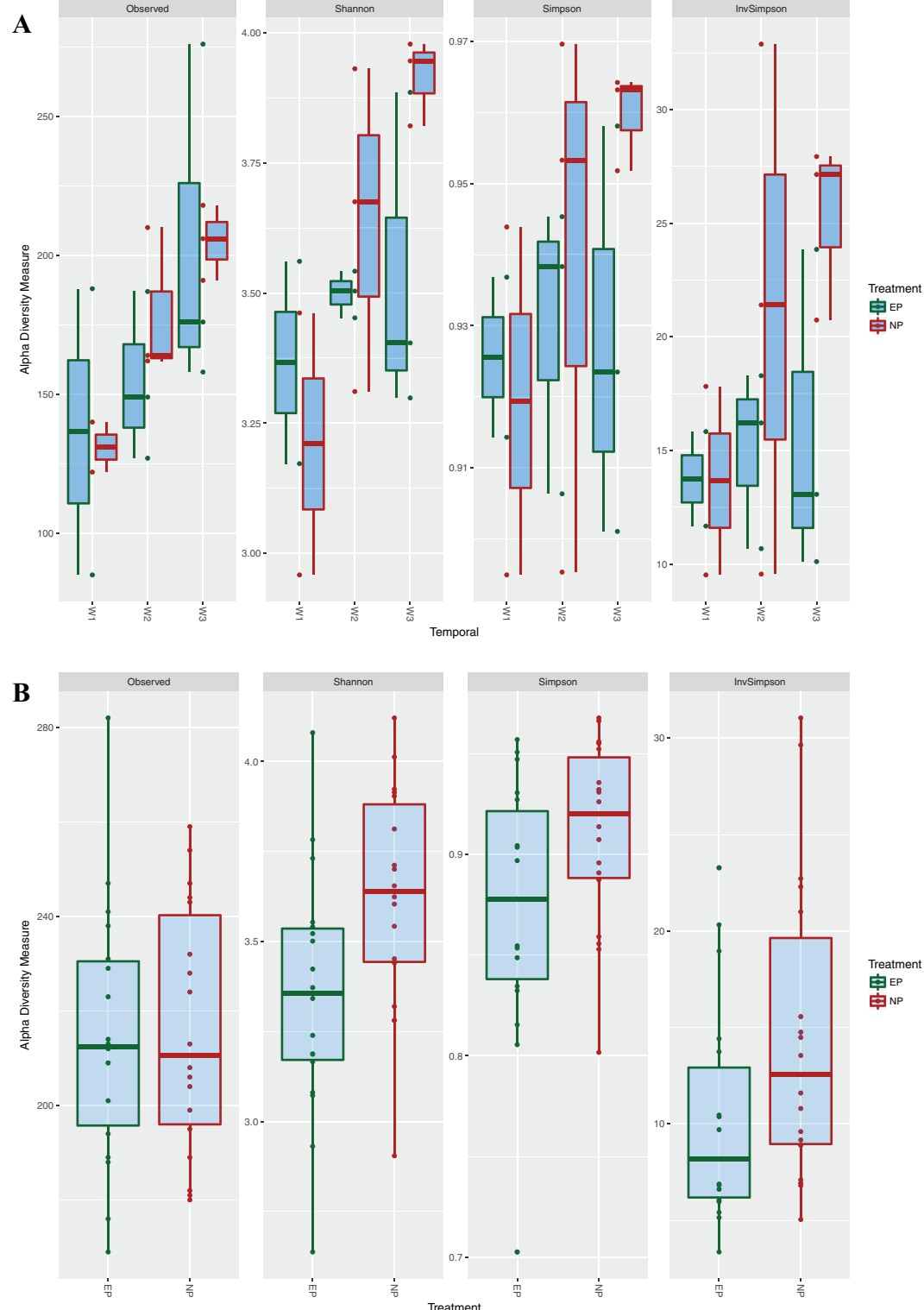

**Figure 5 Alpha diversity measures among photoperiod regimes.** Alpha diversity measures for the two different photoperiods, normal (NP) (12L:12D) and extended (EP) (23L:1D). A shows boxplots of α diversity during the entrainment period (first 3 weeks), divided by each week. B shows boxplots of α diversity estimates from samples taken during the circadian experiment.               

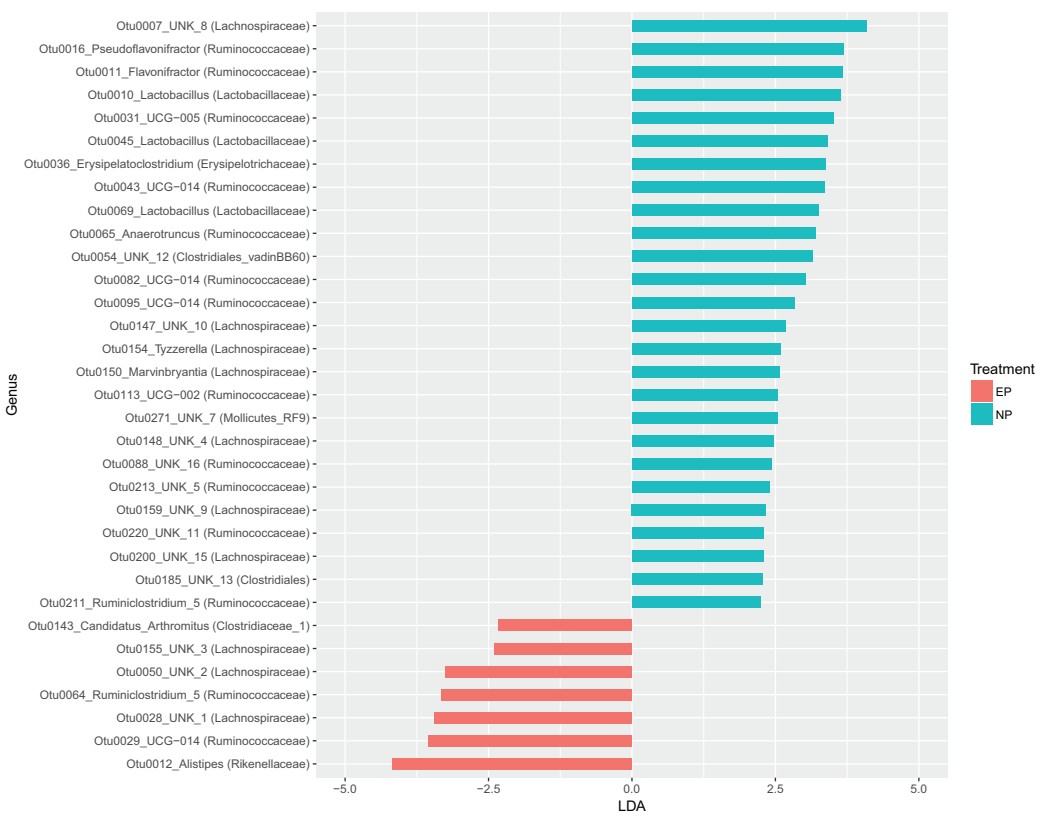

**Figure 6** **LDA plot showing differential enrichment of taxa among photoperiods.** A plot of the results from Linear Discriminant Analysis Effect Size to determine differential enrichment of taxa between photoperiod treatments. Of 33 differentially enriched taxa between treatments, 26 were enriched above an LDA score of 2 the normal photoperiod treatment, whereas the rest were enriched in the extended photoperiod treatment.

## Cecal microbiota oscillations show concordance with host circadian rhythms

Abundance data for 382 OTUs were analyzed for circadian oscillations using JTK_cycle. For the NP treatment, five OTUs oscillated with a significant 24-h rhythm, whereas one OTU oscillated with a 36-h rhythm ($P_{adj} < 0.05$) (Table 2). Except for the taxon oscillating on a 36-h period, all other oscillating OTUs had a low phase shift (0–12 h), indicating that abundance of these taxa follows the host rhythms closely. On the other hand, six OTUs were found to oscillate rhythmically in the EP treatment. Three of these were on 24-h rhythm, and three were in a 36-h rhythm ($P_{adj} < 0.05$) (Table 3). However, all the oscillating OTUs in the EP treatment showed prolonged phase-shifts, ranging from 15 to 33 h.

## Fecal microbiota is not reflective of cecal microbiota

The large majority of OTUs found in the cecal and fecal samples belonged to the phylum *Firmicutes*, followed by *Bacteroidetes* (data not shown). These two phyla are commonly found in the cecal chicken microbiome (*Oakley et al., 2014*). However, at the family level, there were distinct differences between cecal and fecal samples. The cecal samples (Day 4–20) were mainly composed of *Ruminococcaceae* (ca. 50–75%), followed by

**Table 2 Taxa that showed rhythmic oscillations in birds raised in 12/12 LD treatment.**

| Taxa | Adjusted p-value | Period | Phase shift | Amplitude |
|---|---|---|---|---|
| Firmicutes, Clostridia, Clostridiales, Defluviitaleaceae, Defluviitaleaceae_UCG-011 | 0.0005 | 24 | 0 | 0.0005 |
| Firmicutes, Clostridia, Clostridiales, Ruminococcaceae, Oscillibacter | 0.0142 | 36 | 33 | 0.0016 |
| Firmicutes, Clostridia, Clostridiales, Ruminococcaceae, Ruminococcaceae_UCG-014 | 0.0196 | 24 | 12 | 0.0007 |
| Firmicutes, Clostridia, Clostridiales, Ruminococcaceae, Ruminococcaceae_UCG-014 | 0.0312 | 24 | 0 | 0.0001 |
| Firmicutes, Clostridia, Clostridiales, Lachnospiraceae, NA | 0.0358 | 24 | 3 | 0.0021 |
| Firmicutes, Clostridia, Clostridiales, Ruminococcaceae, Anaerotruncus | 0.0417 | 24 | 0 | 0.0001 |

Note:
Oscillating cecal microbiota members in the normal photoperiod (12L:12D) treatment.

**Table 3 Taxa that were oscillating with a rhythm in birds raised under 23/1 LD treatment.**

| Taxa | Adjusted p-value | Period | Phase shift | Amplitude |
|---|---|---|---|---|
| Firmicutes, Clostridia, Clostridiales, Christensenellaceae, Christensenellaceae_R-7_group | 0.0043 | 24 | 21 | 0.0006 |
| Firmicutes, Clostridia, Clostridiales, Lachnospiraceae, NA | 0.0073 | 24 | 15 | 0.0007 |
| Firmicutes, Clostridia, Clostridiales, Ruminococcaceae, Ruminococcaceae_UCG-004 | 0.0142 | 36 | 21 | 0.0005 |
| Firmicutes, Clostridia, Clostridiales, Lachnospiraceae, NA | 0.0266 | 36 | 33 | 0.0023 |
| Firmicutes, Clostridia, Clostridiales, Ruminococcaceae, Ruminococcus_1 | 0.0417 | 36 | 24 | 0.0024 |
| Firmicutes, Clostridia, Clostridiales, Ruminococcaceae, Ruminiclostridium_5 | 0.0417 | 24 | 21 | 0.0005 |

Note:
Oscillating cecal microbiota members in the extended photoperiod (23L:1D) treatment.

*Lachnospiraceae* (ca. 20–40%). On the other hand, the fecal samples (Day 16–20) were largely composed of *Lactobacillaceae* (ca. 10–75%), followed by *Ruminococcaceae* (ca. 50%), *Clostridiaceae_1* (ca. 25–60%) and *Lachnospiraceae* (ca. 5–20%). The cecal samples from the entrainment period (days 4–18) group together closely with the circadian cecal samples (day 19–21), and show a temporal movement as chicks gets older.

Principal coordinates analysis shows a clustering of the three different sample types (Fig. 7), with overlap between the cecal microbiota as noted previously. The fecal microbiota is furthest removed from the two cecal populations, whereas the two cecal populations (CC = Day 19–20, EC = Day 4–18) start out further apart and converge with the passage of time (and chick age). The PERMANOVA results indicate that these three populations do not have the same centroid and are significantly different from each other ($P = 0.001$, 999 permutations). Weighted and unweighted UniFrac analyses also showed these communities to be significantly different ($P < 0.001$).

## DISCUSSION

### Expression of clock genes in the brain and ceca for the two photoperiods

Circadian gene expression oscillation patterns found in this study were in line with what has been previously reported about photoperiods and rhythmic oscillations in various

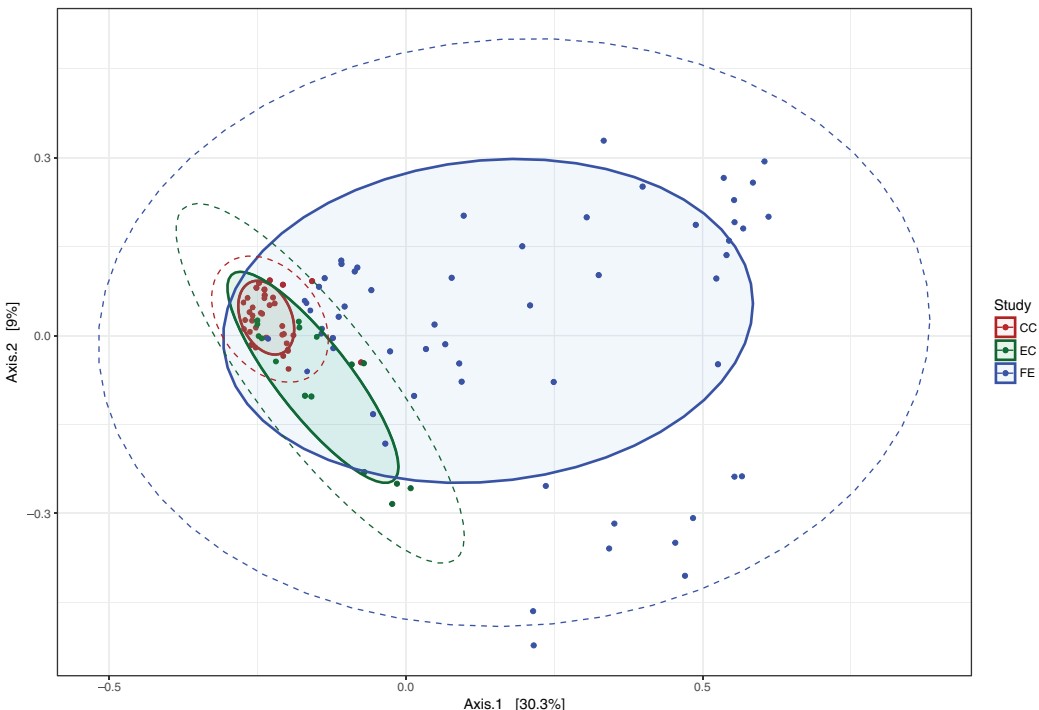

**Figure 7** **PCoA Plot of early life cecal and fecal microbiota.** Principal Coordinate Analysis plot of cecal and fecal bacterial communities in chicks during early life (microbiota acquisition period). CC = cecal samples Day 19–20, EC = cecal samples Day 4–18, FE = fecal samples Day 16–20.

vertebrates including chicken. Particularly, these results agree with *Abraham, Albrecht & Brandstatter (2003)* and *Turkowska et al., 2014*, both of which studied circadian gene expression in the brain of sparrows and chickens, respectively. This study confirms that chicks entrained to the NP (12/12 LD) have a functioning circadian rhythm in both the brain and the ceca, whereas chicks entrained to the EP (23/1 LD) do not show a functioning circadian rhythm in the brain or the ceca. In essence, the chicks entrained to the EP could be said to be in a constant state of phase shift, akin to jetlag experienced by people.

## Different photoperiods promote different microbiota membership and structure

While two birds per time point is low for studies to characterize differences among circadian time-points, the sampling design in this study was focused on characterizing microbiota community profiles between treatment groups. Although the underlying circadian gene expression profiles under different photoperiods have been previously demonstrated in birds and chicken, in this study we generated these profiles to confirm these reported phenomena. However, additional replicates per time point would be valuable and crucial for querying microbiota differences between circadian time points.

Various analysis of α and β diversity showed that the cecal microbiota differed significantly between the two photoperiods. Overall, NP treatments supported significantly

greater α diversity over both the entrainment period and the circadian sampling. Examining the unique genera more closely revealed that the chicks entrained to NP possess genera that are typically associated with healthy guts, whereas the chicks entrained to the EP possess genera that are typically found in diseased guts. The most abundant genus represented in both photoperiods was the *Faecalibacterium*, which belongs to the class *Clostridia* and the phylum *Firmicutes* and is considered a common gut microbe in chickens (*Oakley et al., 2014*). Similarly high proportion (>90%) of *Firmicutes* has been reported in 21-day old layer chicks (*Kers et al., 2018*).

While the microbial communities acquired under the two photoperiods were found to be different according to the diversity metrics, the presence and enrichment of specific taxa under each treatment is perhaps more biologically relevant and interesting to the poultry industry. Analysis of differential enrichment showed a lopsided distribution of enriched taxa between the two treatments. The genus *Alistipes*, which was only found in the EP and belongs to the family *Rikenellaceae*, thrives on high-fat diets and grows especially well in the gut of people suffering from obesity (*Clarke et al., 2013*). Furthermore, it has been found in higher numbers in patients suffering from Irritable Bowel Syndrome (*Saulnier et al., 2011*) and children with Autism Spectrum Disorder (*De Angelis et al., 2013*). Two other enriched taxa (out of seven enriched in EP) were *Ruminiclostridium* and *Blautia.* The enrichment of *Blautia* spp (family *Lachnospiraceae*) has been reported in patients with primary sclerosing choalangitis (*Torres et al., 2016*), a chronic liver disease with links to inflammatory bowel disease. *Ruminiclostridium* (family: *Ruminococcaceae*) has been found to be important in the metabolism of lignocellulosic biomass (*Sheng et al., 2016*), which is a component of plant-based protein and energy sources (corn, soy). The enrichment of this taxon suggests a functional shift to optimize energy utilization from plant-based feed. Altogether, differential enrichment of specific taxa in EP suggest an early shift in energy metabolism profiles, and perhaps point to the origins of metabolic disorders in birds raised under industry standard photoperiods.

Conversely, taxa enriched in the NP treatments were also suggestive of associations to metabolic health. The family *Christensenellaceae*, which was found at a higher abundance in the gastrointestinal tract of chicks entrained to NP, has been associated with a reduction in body weight and adiposity in mice. It has been found in higher numbers in the gut microbiota of people with a lower body mass index and has been shown to have a strong protective effect against visceral fat (*Goodrich et al., 2014*). *Eubacterium hallii*, a common gut microbe with an important role in maintaining intestinal metabolic balance, was also found at a higher abundance in the gut microbiome of birds entrained to the NP compared to the EP. This gut microbe is able to utilize glucose and the fermentation intermediates acetate and lactate. Lactate accumulation has been associated with malabsorption and intestinal diseases (*Engels et al., 2016*). Finally, three *Lactobacillus* members were found to be enriched in the NP treatment (LEfSe analysis). *Lactobacillus* spp are a well-studied group with various known benefits for metabolic and gut health, from antimicrobial activity (*Schillinger & Lucke, 1989*; *Silva et al., 1987*), to their probiotic activity (*Marco et al., 2017*; *Patten & Laws, 2015*). Furthermore, the
enrichment of *Lactobacillus* in the NP treatment suggests positive implications for gut health in the context of pathogen exclusion. For example, *Shaufi et al. (2015)* reported depletion of *Lactobacillus* when pathogenic bacteria like *Clostridium* were enriched. While the mechanisms for selective colonization of specific, beneficial microbes need to be further investigated and understood, our results provide a framework for relating normal circadian activity in early life to gut health. The enrichment of beneficial gut microbiota in NPs can potentially become an inexpensive approach to improve gut health in poultry.

The results show that cecal microbiota acquisition starts diverging (based on α diversity) as early as the first week in birds raised under different photoperiods. As these differences are observed when the only variable was photoperiod suggests that rhythmic physiological processes (as inferred from clock gene expression) may directly influence the colonization efficiency of different microorganisms. A secondary possibility is that the EP affects feeding behaviors and patterns, which are also likely to directly influence the acquisition and colonization process. This study did not measure feed intake specifically, and resolving that association was beyond the scope of this study. Specifically, as poultry rearing systems all utilize ad libitum feeding, our intention was to assess only the effect of photoperiods on circadian rhythms. However, we did observe that birds in 12/12 LD did not entirely stop feeding during dark hours, and also that birds in 23/1 LD did not constantly feed during all hours. We also found that the final weights of birds raised in either photoperiod were not significantly different, but as this study did not systematically track performance data, the source or implication of this finding is unclear.

Despite the significant differences in α and β diversity, the PCoA plots show overlap between the NP and EP microbial communities, which is not entirely unexpected given the same tissue source, age, and diet of the subjects. Overall, the differences observed in microbiota communities, and the clear observation of early and rapid differentiation of microbiota communities within the first week of life emphasize the potential utility of using photoperiods to modulate gut microbiota structure and function.

## Cecal microbiota oscillations

On the one hand, we found that five OTUs in the NP treatment oscillated in a 24-h rhythm in synchrony with their host. On the other hand, cecal gut microbiota members in the EP did not oscillate in a 24-h rhythm and were not in synchrony with their host. In addition, they exhibited greater phase shifts, further indicating the absence of rhythmic oscillations. While mammalian studies (*Thaiss et al., 2014*) have shown strong signals of gut microbiota oscillations in synchrony with the host circadian clock, our study did not show a comparable fraction of oscillating microbiota. Mouse studies have showed that these oscillations represent both compositional and functional differences of the microbiota (*Wu et al., 2018*), and the same processes are likely in chicken. However, a relatively small number of taxa, representing a small fraction of the gut microbiota, were found to be oscillating. One potential explanation for this pattern is that the birds used in our study were placed on ad libitum feed, whereas mammalian studies typically use time-restricted feeding. It has been shown that gut microbiota oscillations are responsive

to the host circadian rhythm, as well as feeding times (*Adamovich et al., 2014*; *Asher & Sassone-Corsi, 2015*; *Hatori et al., 2012*).

Overall, the results showed that a small fraction of the total cecal microbiota oscillated with a significant detectable rhythm (based on JTK_Cycle) in either photoperiod treatment, and fewer still oscillated with a 24-h rhythm. When taxa with significant 24-h rhythms were found, they were almost exclusively in the NP treatment. The absence of 24-h rhythms and protracted phase shifts observed in the EP correspond with the host circadian gene expression, which showed a complete lack of 24-h rhythms, especially in the cecal tissue.

One of the potential caveats in this study is the lower replication of microbiota sampling, in comparison to mice studies which have previously reported on these phenomena. For example, *Thaiss et al. (2016*, *2014)* used 5–10 replicates per time point, compared to two replicates in this study. However, one major difference between mice and chicken studies is the suitability of fecal samples for gut microbiota studies. While the applicability of mouse data for human health has been discussed (*Nguyen et al., 2015*), mouse fecal pellets are an accepted and reliable source of information about gut microbiota. However, chicken fecal samples are not a reliable indicator of gastrointestinal tract microbial communities as reported previously (*Stanley et al., 2015*) and confirmed here (in early life as well). Together with the suitability of fecal samples, and the smaller space requirements, longitudinal and temporal studies with higher replication is less challenging in mouse models compared to chicken models. While our study provides initial evidence of the association between host circadian and gut microbiota oscillations in chicken, further confirmation of mechanisms and functional outcomes will require additional data. Future studies would benefit from use of novel, non-invasive approaches to assay gut microbiota in chicken and other avian models, which currently have to rely on invasive sampling.

## Cecal vs. fecal microbiota communities

This study showed that fecal and cecal microbiota communities are significantly different even in early life, during the microbiota acquisition period. Furthermore, we also found that these differences do not follow any discernible pattern during the acquisition period (first 3 weeks) or later. While overlap in the cecal and fecal communities was observed, and they are in broad agreement with the findings of *Stanley et al. (2015)* and *Oakley & Kogut (2016)*, this data shows that fecal samples are not a reliable indicator of divergence in gut microbiota colonization, membership, or structure. Therefore, our data shows the unsuitability of the fecal microbiota as a surrogate for early life cecal microbiota, limiting its utility as a tool in longitudinal studies of the same individuals. While the fecal data provided a broad snapshot of each treatment group, the fecal data was not tracked at the individual level, making it impossible to correlate with individual cecal data.

## CONCLUSIONS

Here, we present the first report on avian circadian and related gut microbiota oscillations, comparing the consequences of NP vs. EP exposure. This study is also the first to describe differential microbiota acquisition under different photoperiod regimens in birds,

or in any vertebrates to our knowledge. Comparison of fecal and cecal microbiota in early life showed that fecal microbiota is not a reliable indicator of early life colonization. This study provides evidence for a framework linking photoperiod-driven circadian rhythms in early life to benefits for gut health. While this study provides the first evidence of these associations in early life, additional investigations of similar and variable photoperiod regimens and their influence on microbiota are required. Additionally, in-depth understanding of the mechanisms of selective microbiota colonization under photoperiods, their functional importance, and the later-life benefits for the host are required to make this knowledge applicable for animal and human health. Finally, this study points to potential applications for the modulation of colonization by beneficial microbiota in livestock species, especially in the context of raising antibiotic-free animals.

## ACKNOWLEDGEMENTS

We thank Hoa Nguyen-Phuc, Ralf Singh-Bischofberger, and Rohit Rohra for assistance with experimental sampling. We also thank the Texas A&M University Poultry Research Center for logistical help in the performance of this study.

### Funding

The authors received no funding for this work.

### Competing Interests

The authors declare that they have no competing interests.

### Author Contributions

- Anne-Sophie Charlotte Hieke performed the experiments, analyzed the data, prepared figures and/or tables, authored or reviewed drafts of the paper, approved the final draft.
- Shawna Marie Hubert performed the experiments, authored or reviewed drafts of the paper, approved the final draft.
- Giridhar Athrey conceived and designed the experiments, performed the experiments, analyzed the data, contributed reagents/materials/analysis tools, prepared figures and/or tables, authored or reviewed drafts of the paper, approved the final draft.

### Animal Ethics

The following information was supplied relating to ethical approvals (i.e., approving body and any reference numbers):

Texas A&M University's Institutional Animal Care and Use Committee provided approval for this research (IACUC 2016-0064).

### Data Availability

Athrey, Giri (2018): circadian.tar.gz. figshare. Dataset.
DOI 10.6084/m9.figshare.6938249.v1.

## Supplemental Information

Supplemental information for this article can be found online at http://dx.doi.org/10.7717/peerj.6592#supplemental-information.

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
