# Peer review of "Circadian disruption and divergent microbiota acquisition under extended photoperiod regimens in chicken"

_PeerJ, doi:10.7717/peerj.6592_

## Round 0.1 · original submission · Major Revisions

I have reviewed your appeal on the decision made to reject your manuscript. The major concern was that there was too little replication in your study. However, you have made clear in your appeal letter that 18 birds were used per treatment. You have also acknowledged that two replicates per time point is not ideal for assays of circadian oscillations, but have pointed out demonstration of such oscillations in birds was not the novel finding here and only included to confirm a well-established phenomenon. In light of these points and other comments made to the Editor in your appeal, I am recommending your manuscript undergo major revisions. When revising your manuscript please make it really clear how many birds were used at each time point and add a statement regarding the low number of time points included in the study and how additional replicates would facilitate future research. Please note that your revised manuscript will be sent out for review again.

· Appeal

Appeal

Dear Drs. Hoyles and Souza

I want to thank you and the reviewers for their thorough and quick review of our manuscript. I am writing you to formally request that you reconsider the earlier decision of ‘Reject’ and change to ‘Major Revision’. We agree with most of the points and suggestions made by the reviewers, but disagree with the primary criticism regarding the low replication. Our reasoning is given in more detail below:

1. Our primary objective in the study was to assay microbiota differences in birds raised under different photoperiods. For that question, we base our cecal community comparisons on data from 18 individuals per treatment. This is a reasonable number of birds for this type of comparison based on similar literature, and on power analyses.

2. We agree that 2 replicates per time points is not ideal to assay circadian oscillations. However, the demonstration of circadian oscillation in birds raised under different photoperiods is NOT one of the novel findings here. Extensive literature on circadian biology has already demonstrated that phenomenon. In our case, we have nonetheless generated this data for confirmatory purposes - to clearly show that the expected circadian patterns are indeed occurring in these birds. In the next version, we can make it clear that the time series data is presented here purely for confirmatory purposes of a well established phenomenon.

3. We are confident that we can address the remaining comments made by the reviewers. For example it has been previously demonstrated that fecal and cecal microbiota are not reflective of each other in chicken. This is not a new finding here. We only present it in the specific context of early life. However, if this section is not seen as highly relevant to our overall goals, this is a part we can leave out of the next draft. We are pleased to see that all the reviewers made references to the thorough and repeatable description of our methods, except for the concern about replication.

4. Many of the other comments are requesting additional details/plots which we are able to generate and satisfy the reviewers.

Therefore, I request you to kindly give us an opportunity to resubmit a revised manuscript. As a new investigator, with young trainees in my lab, I would greatly appreciate being able to continue within this review process, instead of a new prolonged review at other journals.

best,

Giri


· · Academic Editor

Reject

Your article has been reviewed by three experts. While one reviewer was supportive of your study, two raised concerns over the number of samples (and hence the power) of the study and specific aspects of the article (please refer to the annotated PDF kindly provided by one of the reviewers – it is best viewed using Adobe Reader; Preview on a Mac does not allow all comments to be viewed).

Full information should be provided for the qPCR (including primer sequences and control housekeeping genes). In addition, it is good practice to include negative controls in 16S rRNA gene sequence studies to account for kitome/reagent contaminants. Non-inclusion of a negative control should be justified in Methods.

Microbiota literature referenced should focus solely on the chicken microbiota, not human nor mice – avian and mammalian microbiota composition and, thereby, function are very different from one another, and comparisons cannot be made beyond generic statements such as the microbiota is intimately linked with host health. All reviewers agreed your study is interesting and could be relevant to the poultry industry, though its potential relevance is not mentioned in the article’s conclusions.

Given the shortcomings of the work I am afraid that it is not acceptable for publication in PeerJ and so must be rejected. If you are able to address the reviewers’ comments in the future, I would encourage resubmission of an appropriately revised manuscript with properly formatted figures.

# Reviewer 1 ·

Basic reporting

The manuscript is well written, but one sentence "This study is also the first to report differential microbiota acquisition under different photoperiod regimes" in the abstract should be modified since this report is not the first paper to disclose this phenomenon.

Experimental design

Overall is good. But the last part for the comparison between cecal content and feces is somewhat unrelavant with other parts. The authors should address this point.

Validity of the findings

Data is clear, but I strongly recommend the authors to add new parameters comparison for PCoA to further reveal the difference between N and E (e.g. the PC1 distance for each sample in these 2 groups). The current data is not clear enough to show the overall difference between N and E.

Additional comments

I think this paper is potential interesting but the authors should address all the above points before publishing.

·

Basic reporting

The English is clear. The article is overall well written.
The Background is sufficient and maybe slightly long regarding the introduction. The references are appropriates.
The structure of the article is standard and professional.
Most results presented are of significance to the hypothesis.

Experimental design

The paper reports original primary research within aims and scope of the journal. The research questions is well defined and meaningful with investigation performed to a high technical and ethical standard. Methods were described with sufficient detail and information to replicate. However, the number of samples taken at each time point (n =2) is too low to ensure the validity of the results.

Validity of the findings

The validity of the results is difficult to assess due to the low number of replicates although rigorous methods are applied.

Additional comments

The paper describe the effect of light time exposure on chicken circadian rhythm and its effect on the cecal microbiome. This is a very interesting study design and the questions raised by the paper are of real interest to the scientific community. However, the authors are encouraged to at least double the number of samples taken at each time point to be able to draw any conclusion from the results and assess the significance of their findings.

·

Basic reporting

The structure is as required for submission to PeerJ. The subheadings for the discussion are not needed. The introduction is clear and thorough although there are some vague statements that need clarification (line 136 and 149). The addition of reference to previous studies on healthy chicken gut microbiota would strengthen the introduction.
There are a number of sentences in the results section that should be moved (if not already stated) from the results to the methods or discussion section (see annotated PDF), as they give details of the methodology or discuss what the results may mean (respectively).
A number of figures need formatting so that the results are clearer to the reader. Figure 1 has axis labels and splitting the figure into and A, B, C and D (per row) would clarify what photoperiod or sample is being shown. There is no clarification of acronyms used for samples in Figure 2. The lefse output in figure 5 is only shown at family level and a number of families are repeated more than once. Would it not be more informative to show the lefse output with all bacterial groups shown at all phylogenetic levels of classification? There is no rationale in the methods why only family is shown in this figure.

Experimental design

This study is within and the design aims to answer a novel question that has not been explored before. However, aim D states “to determine if fecal microbiota is representative of cecal microbiota” (in the abstract). This is not novel as this previous publications have commented on the similarity of fecal and cecal microbiota representing the cecal microbiota eg. Stanley et al. 2015 BMC Microbiology (DOI 10.1186/s12866-015-0388-6). There is a reference to this work already being published on line 549, if this question does need re-addressing then justification for this should be added to the manuscript. When this aim is stated later in the manuscript, in the introduction, “early life cecal microbiota” is mentioned. If this aim D is to assess whether fecal microbiota is representative of cecal microbiota, solely in young birds, this needs to be clarified and justification to why this may be different to adult birds added.
The paragraph in the methods describing the QPCR undertaken (line 225) does not mention any use of housekeeping genes as controls or state the sequence of the primers used.

Validity of the findings

In the results section it is not clear what defines if oscillation is present or absent in the clock genes studied using QPCR. Statistical testing should be added to show that the lack of oscillation in Per2 is statistically significant.
There is scope for more comparison between the results here and previous chicken microbiota papers in the discussion section. Comparison between the finding here in chickens subjected to different photoperiods and normal or diseased chickens (for example Shaufi et al. 2015 Gut Pathogens and Wei et al. 2013 Poultry Science) would strengthen the discussion section and give better insight into the potential implications of these findings to the poultry industry. In the paragraph beginning line 481, but as this is more a reflection of the normal light periods chickens would be exposed to with no artificial light would it not be more useful to compare the results of the normal photoperiod to the cecal and fecal microbiota previously reported in disease free chickens? The importance of the poultry industry and the reasoning behind different photoperiods in the industry were included in the introduction section. However, this had not been commented on in the discussion. Suggestions on how the results reported here may affect the growth and/or egg laying of chickens and any potential changes that would be needed to be made to current photoperiod regimes would strengthen the discussion.

Additional comments

There is the need for clarification of which photoperiod and/or sample type is being described in a number of sentences in the results section (lines 302 and 340).
The inclusion of figure 2 but at phyla, class, order and family level to supplementary material would give a more complete overview of the fecal and cecal profiles of the samples analysed.

---

## Round 0.2 · Minor Revisions

Thank you for taking on board Dr Leng's extensive feedback. Your revised manuscript has been reviewed by two experts, both of whom are requesting minor revisions be made before the manuscript can be accepted for publication. Please address all the reviewers' comments when preparing your revised manuscript.

On behalf of one of the reviewers I am providing the following feedback you may wish to consider for future studies in this area. Provide data regrading the birds' performance in terms of bird weight, feed intake and feed conversion ratio. For future studies to have a significant impact on the poultry industry, these are critical factors that require investigation.

·

Basic reporting

No comment

Experimental design

Information on the QPCR methods have been added as requested. Could the primers used be added in a table to supplementary items? Clarity on the novelty of the aim assessing the differences between fecal and cecal microbiota has been added. The final conclusions paragraph (line 580) be used to emphasise how this part of your study is different to the references mentioned.

Validity of the findings

Paragraphs beginning lines 483 and 500 discuss the differences in microbiota seen between the two different phot regimes. These paragraphs include lots of references that have found similar findings to the results of the bacterial community profiling of the two groups of chickens. Could you end each of these paragraphs with a sentence linking this directly back to your study.

Additional comments

The authors have taken on my suggestions and there has been great improvement in the manuscript since the first draft submitted for review. The figures are much clearer and the results section is easier to follow and flows better. I have a few minor suggestions that can be found below a few comments. I believe that if these are addressed the manuscript will be ready for publication.
Line 220: Could you add the primer sequences in a table to supplementary items?
Line 356: Could a figure be added to the supplementary items to complement this description of dominant phyla in the two photoperiods.
Lines 377 and 401: are these references to Figure 5, specific to part A or B?
Line 430: Could a figure be added to supplementary items to illustrate the comparison of fecal and cecal microbiota mentioned here?
Paragraph beginning line 472: This paragraph discusses bacterial diversity and community profiles, could this paragraph solely concentrate on bacterial diversity and include differences in alpha diversity seen.
Line 540: state what these changes you found were to remind the reader of what is being discussed.

·

Basic reporting

Generally clear and professionally written manuscript.

As the study focuses on the early life of birds, it is important that the age of the birds is mentioned in the abstract.

From reading the introduction, I made the assumption that the study was performed on broiler chickens; It is not until the start of the materials and methods that I realised this was performed on layer hens. Can it be clearer in the abstract (and title) that this was done in newly hatched laying hens to better inform the reader from the start.

Some inconsistencies in materials and methods on reporting manufacturer of reagents used, for example line 208: "RNAlater" (no further details), vs line 236-237: "Q5 High-Fidelity DNA polymerase (NEBNext High-Fidelity 2X PCR Master Mix, New England BioLabs, Ipswich, MA"

Fig. 1 and Fig. 2 legends: "Error bars are standard errors" - but this is a box-plot.

Good quality of tables and figures presented, although Table 1 would be better placed as supplementary materials.

Line 443: the term "flora" should be avoided.

Some inconsistencies in abbreviation used NP/EP

Experimental design

Original piece of research addressing gap in knowledge.

The study aims to answer a number of relevant scientific questions to the industry.

The main concern lies over the number of replicates with only 2 cages used per treatment (20 birds per cage). It seems that birds were considered as the replicate instead of the cage. Co-housed birds would not be statistically independent, and it could be argued that the effect observed is due to cage or room effect rather than the photoperiod effect.

Good level of details provided in materials and methods.

Line 184: As a strong influencing factor on the microbiome, and to enable comparison with future studies (or replication): basic details of the feed used should be stated.

Line 186; "following the producer manual", this is too unclear.

Line 235: as the gold standard practice, was a bead-beating step used prior to DNA extraction?

For sequencing: did you use a mock community standard and a negative blank (from the DNA extraction)?

Validity of the findings

Interesting findings, well reported and discussed in the context of existing literature.

Level of Firmicutes reported is very high (90-94%), is this comparable to similar studies for birds this age?

Line 461: the term "jetlag" could be misinterpreted by reader and should be defined more clearly

Line 532-534: "final weights of birds raised in either photoperiod were not significantly different, which shows that the differences observed in microbiota composition was not due to differences in feeding behaviors." this cannot be concluded from bird weight alone as feed intake was not recorded or observed.

A major limitation of the comparison between faecal vs caecal microbiota is related to the sample collection method used where it is not possible to identify which faeces corresponds to each individual bird from a tray collection. This should be also addressed in the discussion section.

Additional comments

It is a real shame that no data regrading the bird's performance in terms of bird weight, feed intake and feed conversion ratio were reported in this study (weight was briefly mentioned towards the end of the discussion). In order for this study to have a significant impact in the industry, this would have been a critical factor to investigate.
I understand this was beyond the scope of the current studies, but a suggestion for future studies on a larger scale.

---

## Round 0.3 · Minor Revisions

Thank you for resubmitting your manuscript. I've been through your amendments and rebuttal letter. I note you state a negative control (blank) was included in your study. This information must be included in the Methods, along with how OTUs associated with the negative control were dealt with in this and other samples.

Please also add a statement to the Methods indicating that the sequence data associated with the study are available for download (either via figshare or (ideally) from GenBank; if the latter, please provide the accession number).

Find attached your revised manuscript with minor edits - some for clarity, some for typos. Please incorporate these changes into the final version of your article. I have added a statement regarding the need to consider cage or room effects in future studies. This is an important point, and one which experienced microbiome researchers will criticise the paper for if the issue is not addressed head on in the article. I hope you will agree this needs to be included in the final version of the manuscript.

---

## Round 0.4 · accepted · Accept

Thank you for working with reviewers to get this article to press. I am happy to tell you that your article is now accepted for publication.

For future reference, non-template negative controls need to be included in sequencing libraries even if no band is visible on PCR gels. Refer to PMID:25387460 and PMID:30046175 and use the information detailed within to inform future experimental design.

#